# Finding Causal Impacts of Social Determinants of Mental Health on Opioid Use Disorder from Clinical Notes

**Madhavi Pagare**
*University of Massachusetts Lowell*, MA, USA
madhavisuyog_pagare@student.uml.edu
**Atqiya Munawara Mahi**
*University of Massachusetts Lowell*, MA, USA
atqiyamunawara_mahi@student.uml.edu

**Inyene Essien Aleksi**
*Merrimack College, North Andover*, MA, USA
essienaleksi@Merrimack.edu
**Mohammad Arif Ul Alam**
*University of Massachusetts Lowell*, MA, USA
*University of Massachusetts Chan Medical School*, USA
*National Institute of Health*, MD, USA
mohammadariful_alam@uml.edu

*Abstract*—Opioid Use Disorder (OUD) affects physical health, mental well-being, and socio-economic stability, underscoring the need to understand the causal effects of social determinants on OUD outcomes. Despite extensive correlation studies between social determinants of mental health (SDMHs) and OUD, specific causal determinants remain unidentified due to the lack of robust causal models. This paper proposes a two-step causal effects identification framework to detect and identify the effects of SDMHs on OUD progression. Firstly, we developed a Multitask Multilabel Clinical-Longformer(MMCL) model to detect social determinants of Mental health from clinical notes, effectively processing and identifying relevant SDMHs within unstructured text data. Secondly, we employed a novel Siamese Neural Network(SNN)-based subgroup discovery technique to ascertain the causal effects of these social determinants on OUD. This technique leverages the Siamese architecture's capability to handle complex relationships and identify homogeneous subgroups within the data, enhancing the precision of causal inference. To support this research, we collaborated with experts to create a new dataset, SDMH-OUD-Clinic, comprising social determinants of Mental health-annotated clinical notes and OUD annotations, sub-sampled from the MIMIC-IV dataset. We evaluated the proposed models using the Infant Health and Development Program (IHDP) dataset and applied them to our newly created SDMH-OUD-Clinic dataset. The results demonstrate the model's effectiveness and provide detailed explanations of the identified causal relationships.

*Index Terms*—Opioid Use Disorder, Causal Effect, Siamese Neural Network, Sub-group discovery, Social Determinants

## I. INTRODUCTION

Social Determinants of Mental Health (SDMHs) significantly influence susceptibility to Opioid Use Disorder (OUD) and its outcomes [1], [2], [3]. These factors shape OUD patterns, but their relationship with OUD risk remains underexplored due to the high costs of clinical trials [4], [5]. Utilizing large-scale observational data and advanced statistical methods, this study simulates a clinical trial to identify the harmful effects of these determinants on OUD. This approach offers a comprehensive framework to understand how specific factors contribute to OUD, guiding targeted strategies to improve healthcare outcomes.

SDMHs operate across individual, community, and societal levels [4], [5], [17], with factors like socioeconomic status, social capital, and criminal justice involvement playing significant roles [4], [5]. Unemployment, unstable housing, criminal justice history, and neighborhood crime contribute to OUD risk, compounded by social isolation and trauma [4], [5]. The complexity of these interactions, combined with fragmented surveillance and healthcare systems, has hindered comprehensive analysis [6]. However, modern statistical methods, including SEM, instrumental variable analysis, and causal forests, enable deeper exploration of these causal effects [7], [8], providing insights that inform targeted interventions and policies.

OUD is a major public health issue affecting individuals, families, and communities, stemming from opioid misuse, including prescription pain relievers, heroin, and synthetic opioids [3], [4], [5]. It leads to severe physical, psychological, and social consequences, contributing to addiction, overdose, and death, while imposing significant burdens on healthcare systems and society [5], [6], [7]. SDMHs, such as socioeconomic status, healthcare access, and community factors, play a critical role in OUD risk. Individuals facing challenges such as unemployment, housing instability, criminal justice involvement, and social isolation increase their vulnerability to developing mental health problems and substance abuse issues, including OUD [7]. SDMHs also impact the effectiveness of OUD interventions [9]. Addressing barriers such as healthcare access, economic hardship, and housing instability is crucial for improving treatment outcomes and harm reduction efforts [2].

Understanding the connections between SDMHs and OUD is essential for creating comprehensive public health strategies that enhance mental health outcomes for individuals with OUD. Research shows that individual, social, and environmental factors shape OUD determinants. Socioeconomic influences, such as educational attainment and rural residency, are associated with OUD across diverse populations [10], [11]. Coexisting conditions like chronic pain and mental health disorders further contribute to OUD vulnerability [12]. Understanding these determinants is crucial for developing effective prevention and intervention strategies. By examining the multifaceted nature of OUD and its determinants, tailored interventions can address the specific needs of diverse populations and mitigate the impact of this complex public health challenge.

In this study, the Multitask Multilabel Clinical-Longformer (MMCL) model and the Siamese Neural Network (SNN) used for causal inference are advanced tools designed to analyze clinical notes and identify SDMHs affecting OUD. While the methods are complex, their practical benefits are clear: the MMCL helps clinicians quickly focus on key SDMHs, and the SNN refines insights by identifying subgroups for targeted interventions. This makes the models' outcomes accessible and actionable for healthcare providers, without requiring deep expertise in machine learning or causal inference.

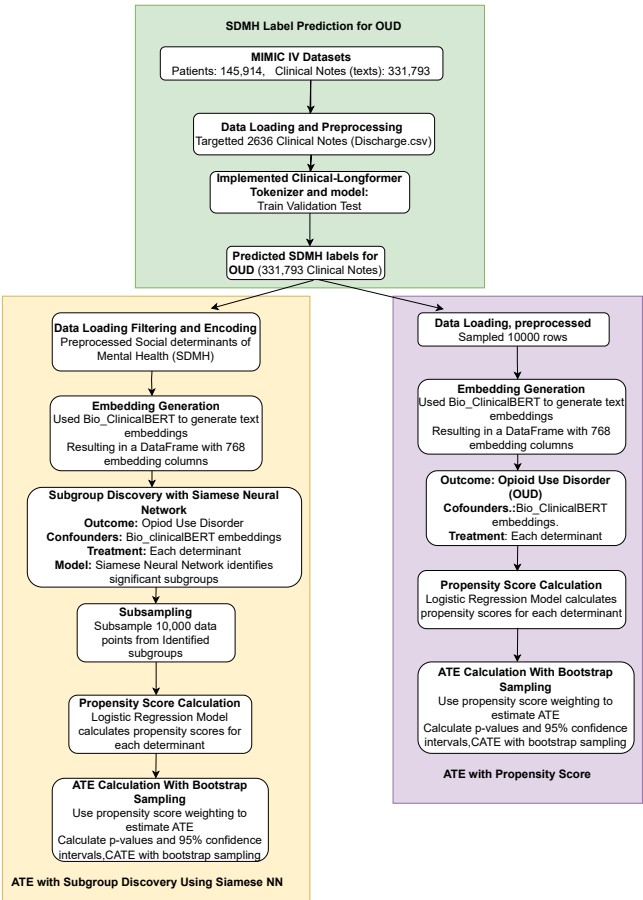

Fig. 1. Overview of Social determinant of Mental Health(SDMHs) prediction for OUD and Causal inference estimates and methods for obtaining them. ATE (Average Treatment Effect); MIMIC IV: Medical Information Mart for Intensive Care IV

## II. METHODS

### A. Overview of Proposed Framework

The proposed framework aims to identify the causal effects of SDMHs on OUD outcomes using a two-step approach. Firstly, a Multitask Multilabel Clinical-Longformer (MMCL) model is used to detect and classify social determinants of mental health from clinical notes. Secondly, a Siamese Neural Network(SNN)-based subgroup discovery technique is employed to determine the causal effects of these social determinants on OUD. This method leverages the Siamese architecture to handle complex relationships and identify homogeneous subgroups within the data, enhancing the precision of causal inference. A new dataset, SDMH-OUD-Clinic, is created for this purpose, and the proposed models are evaluated on both this dataset and the IHDP dataset, demonstrating

their effectiveness in providing detailed explanations of the identified causal relationships (Fig 1).

### B. Multitask Multilabel Clinical-Longformer (MMCL) model for Classifying Social Determinants from Clinical Notes

We developed a Multitask Multilabel clinical text classification framework utilizing the Clinical-Longformer (MMCL) model to efficiently process and categorize lengthy clinical narratives inspired by Mitra et. al. [13] (list of 13 Social Determinants with definition and keywords available in Supplementary Document 1[1].In this algorithm, initially, the model classifies clinical text into categories indicating the presence or absence of OUD by transforming the text into high-dimensional embeddings through Clinical-Longformer and refining these embeddings with a dense layer comprising 64 hidden units and a ReLU activation function. A single neuron with a sigmoid activation function in the output layer computes the probability of OUD. Subsequently, the model further classifies texts based on mental health determinants using a dense layer with 32 hidden units and a sigmoid activation function. If social determinants of mental health (SDMHs) are detected, the model proceeds to identify specific SDMHs out of thirteen possible categories through a multi-label dense layer with 128 hidden units and individual sigmoid functions for each label (Supplementary Document 1[1]). This stacked approach enhances the semantic understanding and nuanced classification of clinical narratives, facilitating deeper insights and more precise categorization.

### C. Subgroup Discovery using Siamese Neural Network(SNN)

#### 1) Subgroup Discovery in Causal Effect

Subgroup discovery in causal effect models involves identifying subsets of a population where the treatment effect is significantly different from the general population, thereby providing more tailored and precise insights into causal relationships. The major challenges in subgroup discovery include handling complex data relationships, preventing overfitting, and accurately estimating treatment effects within subgroups. Our proposed Siamese Neural Network(SNN) [14] architecture addresses these challenges by using a dual-network structure to process treatment and control inputs, leveraging dropout layers to mitigate overfitting, and employing a novel technique to calculate Conditional Average Treatment Effects (CATE) for precise subgroup identification.

#### 2) Problem Formulation

Let $\mathcal{D} = \{(X_i, T_i, Y_i)\}_{i=1}^{n}$ denote a dataset of $n$ individuals, where $X_i \in R^d$ represents the features, $T_i \in \{0, 1\}$ is the treatment indicator, and $Y_i \in R$ is the outcome variable. The objective is to estimate the causal effect of the treatment $T$ on the outcome $Y$. We denote the potential outcomes as $Y_i(0)$ and $Y_i(1)$ for control and treated conditions, respectively. The individual treatment effect (ITE) is defined as:

$$\text{ITE}_i = Y_i(1) - Y_i(0).$$

[1]https://github.com/MadhaviPagare/Causal-Impacts/

The average treatment effect (ATE) is the expected value of ITE over the population:

$$\text{ATE} = E[Y(1) - Y(0)].$$

Subgroup discovery aims to identify a subset $\mathcal{S} \subseteq \mathcal{D}$ where the treatment effect is significantly different from the general population, i.e., finding $\mathcal{S}$ such that the conditional average treatment effect (CATE) is maximized:

$$\text{CATE}_{\mathcal{S}} = E[Y(1) - Y(0) \mid (X, T, Y) \in \mathcal{S}].$$

*3) Model Architecture*

The SNN architecture begins with an input layer, where two types of inputs are provided: the features $X$ and the treatment indicator $T$. These inputs are processed through a shared network that comprises a dense layer with $h$ hidden units activated by a ReLU function, followed by a dropout layer with a dropout probability of $p$. This shared network is designed to learn a common representation of input features. For both the treatment and control groups, the architecture includes a treatment-specific network, each with a dense layer of $h$ hidden units and ReLU activation. Multiple dense layers with 100 units, interspersed with dropout layers, follow, preventing overfitting by randomly setting a fraction of input units to zero during training. The outputs from these networks are then concatenated with the shared representation. The final layer of the SNN is an output layer that comprises a dense layer with a single unit and a linear activation function. This layer is responsible for predicting the outcome variable for both the treatment and control groups.

The SNN for subgroup discovery is formulated as follows:

$$\text{SNN}(X, T) = f_\theta(X) \parallel g_\phi(T),$$

where $f_\theta$ and $g_\phi$ are neural networks parameterized by $\theta$ and $\phi$, respectively. The final prediction is obtained by concatenating the shared representation $f_\theta(X)$ with the treatment-specific representation $g_\phi(T)$ and passing through a dense layer:

$$\hat{Y} = \text{Dense}([f_\theta(X), g_\phi(T)]).$$

The model is optimized using the MSE loss function.

*4) Training SNN Network*

Initially, the dataset is split into treatment and control groups. The SNN is trained on both using the Mean Squared Error (MSE) loss function, which measures the difference between predicted and actual outcomes. After training, the model predicts outcomes for both groups. The conditional average treatment effect (CATE) is then calculated by taking the difference between the predicted outcomes for the treated and control groups within identified subgroups. The subgroup $\mathcal{S}$ with the largest CATE is identified as having the most significant treatment effect.

## III. RESULTS

*A. SDMH-OUD-Clinic Database Generation*

MIMIC-IV is an extensive electronic health record dataset developed by Beth Israel Deaconess Medical Center (BIDMC)

---

**Algorithm 1** Siamese Neural Network(SNN) based Subgroup Discovery

---

1: **Input:** Data $\mathcal{D} = \{(X_i, T_i, Y_i)\}_{i=1}^{n}$, hidden dimension $h$, dropout probability $p$, epochs $e$, threshold $k$
2: **Output:** Identified subgroup $\mathcal{S}$
3: Split $\mathcal{D}$ into treatment group $\mathcal{D}_1$ and control group $\mathcal{D}_0$
4: Create Siamese Neural Network(SNN) model
5: Train SNN with $\mathcal{D}_0$ and $\mathcal{D}_1$ for $e$ epochs
6: Predict outcomes $\hat{Y}_0$ and $\hat{Y}_1$ for control and treatment groups, respectively
7: Calculate conditional average treatment effects $\text{CATE}_i = \hat{Y}_1 - \hat{Y}_0$
8: Identify subgroup $\mathcal{S} = \{(X_i, T_i, Y_i) \mid \text{CATE}_i > k\}$
9: **return** $\mathcal{S}$

---

and Massachusetts Institute of Technology(MIT), with support from the NIH. It contains patient measurements, orders, diagnoses, procedures, treatments, and deidentified free-text clinical notes. The discharge summaries provide detailed overviews of patient hospitalizations, including sections like chief complaint, history of present illness, medical history, hospital course, physical exams, and discharge diagnoses [7]. For our study, we utilized 331,794 anonymized discharge summaries from 145,915 patients treated at BIDMC, with detailed population distribution for each social determinant provided in the Supplementary Document 2[1]. Initially, we focused on 2,600 clinical texts from discharge.csv file to ensure a balance between in-depth analysis and computational efficiency.

We cleaned and standardized the discharge summaries using Python's pandas and regex libraries, removing patient identifiers, extraneous punctuation, and redundant information. Advanced NLP techniques from NLTK, including tokenization, stop word removal, and lemmatization, were applied to refine the text for analysis. Finally, expert guidance was used to annotate the data. Once the dataset was processed, expert annotators manually labeled a subset, followed by the utilization of LLMs via the HLLIA (Human-in-the-Loop-LLM Interaction for Annotation) approach to reduce annotator workload. The process involved selecting a prompt strategy(Supplementary Document 2[1]) with the highest partial correlation coefficient for LLM annotation, guided by clear objectives, iterative testing, and contextual information. This method enhanced the dataset's reliability by ensuring correlations among SDOMH variables.

While an LLM like ChatGPT accelerated the annotation process, inconsistencies in label assignments required manual corrections to ensure accuracy. Additionally, although Chat-GPT could identify missing data, it couldn't resolve it, leaving our experts to apply their domain knowledge to address gaps and ensure the dataset's reliability. As a result, our dataset is comprehensive, capturing key social determinants through thorough preprocessing, expert validation, and iterative refinement. After running experiments using Clinical-Longformer model based on our text classification based SDMHs labels for OUD, we did predictions on 331,794 anonymized discharge summaries to generate our SDMH-OUD-Clinic datasets.

TABLE I
MODEL PERFORMANCE METRICS

| Model | Metric | T1 | T2 | T3 |
|-------|--------|-----|------|------|
| Clinical-Longformer | **Acc** | 99.00 | 95.82 | 96.29 |
| | **F1** | 98.99 | 95.68 | 95.41 |
| | **Prec** | 99.02 | 95.85 | 96.03 |
| | **Recall** | 98.97 | 95.80 | 94.81 |

### B. Evaluation of Multitask Multilabel Social Determinants of Mental Health Classifier

#### 1) Proposed MMCL model for SDMHs prediction

We implemented a Multitask Multilabel Clinical Text Classifier with Clinical-Longformer Embeddings (MMCL) using the Python-based PyTorch platform. The model was trained and tested on approximately 2600 samples (Table I), generating predictions for SDMHs related to OUD.

#### 2) Baseline Model

The Clinical-Longformer model is a domain-enriched language model designed to handle the extensive length of clinical texts, extending the maximum input sequence length from 512 to 4096 tokens[8].Given that the average length of clinical texts is 791.27 words, models like ClinicalBERT [16] face limitations with memory consumption and performance degradation when processing long texts.We implemented traditional models, including RoBERTa, Clinical-BERT, Bio_ClinicalBERT, and Clinical-BigBird, and found that Clinical-Longformer consistently outperformed, demonstrating its superior performance for clinical natural language processing tasks based on accuracy(refer Supplementary Document 2[1]).

#### 3) Predictions on 331,794 Discharge Summaries

Following experiments with the Clinical-Longformer model, we generated predictions on 331,794 anonymized discharge summaries to create our SDMH-OUD-Clinic datasets.

### C. Evaluation of SNN based Causal Effect Model

#### 1) Evaluation on Existing IHDP Dataset

The Infant Health and Development Program (IHDP) dataset is a longitudinal dataset designed to evaluate the effects of early intervention on low birth weight, premature infants [15]. It includes extensive demographic, socio-economic, health, and developmental data collected over multiple years from a diverse cohort of children and their families. Researchers often use the IHDP dataset for causal effect studies to assess early interventions and identify factors that contribute to positive developmental and health outcomes. To evaluate our causal effect model's performance on the IHDP dataset, we use the Precision in Estimating Heterogeneous Effects (PEHE) metric, defined as:

$$\text{PEHE} = \frac{1}{n} \sum_{i=1}^{n} \left( f_1(x_i) - f_0(x_i) - E[Y(1) - Y(0) \mid X = x_i] \right)^2 \quad (1)$$

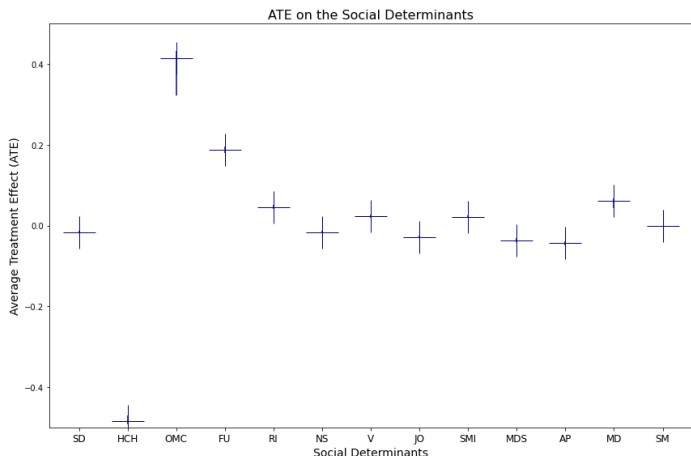

Fig. 2. Average Treatment Effect on Social Determinants of Mental Health (SDMHs) with 95% Confidence Intervals with Siamese Neural Network(SNN)-based Subgroup Discovery

Here, $f_1(\cdot)$ and $f_0(\cdot)$ represent the estimated potential outcomes under treatment and control, respectively. Our SNN-based causal effect model is compared against standard causal inference methods using both in-sample PEHE (computed on training data) and out-of-sample PEHE (computed on held-out test data). The results show that our proposed model consistently outperforms the baselines, especially in higher-dimensional settings like the IHDP dataset (Table II).

TABLE II
PEHE VALUES IN ESTIMATING HETEROGENEOUS EFFECTS. ERROR REPRESENTS 95% CONFIDENCE INTERVAL OF MULTIPLE MONTE CARLO INITIALIZATIONS.

| Method | In-Sample | Out-Sample |
|--------|-----------|------------|
| HEMM-MLP | $1.6 \pm 0.10$ | $1.8 \pm 0.10$ |
| HEMM-Lin | $2.8 \pm 0.32$ | $2.9 \pm 0.33$ |
| Linear-1 | $7.9 \pm 0.46$ | $7.9 \pm 0.47$ |
| Linear-2 | $2.3 \pm 0.18$ | $2.4 \pm 0.21$ |
| k-NN | $3.2 \pm 0.12$ | $4.2 \pm 0.22$ |
| GP | $2.1 \pm 0.11$ | $2.3 \pm 0.14$ |
| CFRF | $2.7 \pm 0.31$ | $3.3 \pm 0.72$ |
| VT-R | $2.5 \pm 0.26$ | $2.9 \pm 0.51$ |
| CEVAE | $2.1 \pm 0.21$ | $2.2 \pm 0.33$ |
| **SNN** | $1.5 \pm 0.20$ | $1.6 \pm 0.25$ |

Here,Table II shows the comparison results with the baseline causal effect algorithms where it is clear that SNN based causal effect model outperforms other baseline algorithms.

### D. Application of SNN Causal Effect Model on Our Opioid Use Disorder Dataset

In our study, we aimed to analyze the causal relationships between specific SDMHs and OUD.

#### 1) Selection of Social Determinants as Treatment Variables

We selected 13 social determinants based on their potential influence on OUD: Socially Detached (SD), Health Care Handover (HCH), Obstacles to Medical Care (OMC), Financial Uncertainty (FU), Residential Instability (RI), Nutritional Shortage (NS), Violence (V), Judicial Obstacles (JO), Substance Misuse (SMI), Mental Disturbance Symptoms (MDS),

Acute Pain (AP), Medical Disability (MD), and Suicide Mortality (SM). Selection was based on literature review and expert consultation, confirming these as significant contributors to OUD.

While the MIMIC-IV dataset does not define the 13 social determinants, we, along with clinical experts, defined them, developed sophisticated algorithms to predict them from MIMIC-IV clinical notes, and validated a portion of the annotations through clinician review.

*2) Opioid Use Disorder (OUD) as Outcome Variable*

The outcome variable, OUD, was defined using clinical diagnoses and self-reported data on OUD and related health issues, aligning with our objective to study the causal effects of social determinants on OUD.

*3) Bio_ClinicalBERT Embeddings as Feature Vectors*

To capture textual and contextual nuances in our dataset, we used Bio_ClinicalBERT to generate text embeddings [16]. Bio_ClinicalBERT, a transformer-based model tailored for biomedical and clinical text, provides strong performance in understanding language in these contexts. The embeddings served as feature vectors in our causal inference analysis, capturing complex relationships between the text data and selected SDMHs, thereby enhancing the robustness of our analysis.

*4) Causal Inference Analysis*

We considered Causal Effect Estimation and Subgroup Discovery Analysis for determinants of OUD, utilizing Bio_ClinicalBERT embeddings and proposed Siamese Neural Networks(SNN). Here are the detailed steps:

1) Data Preprocessing and Embedding Generation: We preprocessed the data from the 'predictions.csv' file by loading it, converting binary values to numeric format, and generating text embeddings using Bio_ClinicalBERT. These embeddings were flattened and incorporated into the dataset for analysis.
2) Subgroup Discovery Using Siamese Neural Networks(SNN): To identify subgroups within the data, we employed a Siamese Neural Network(SNN), which is well-suited for capturing complex relationships and discovering meaningful subgroups with shared characteristics.
3) Subsampling Identified Subgroups: After identifying subgroups, subsampling is applied to create a balanced dataset for detailed analysis, ensuring that the subsequent analysis is conducted on a representative subset of the data.
4) Propensity Score Calculation and Matching: Logistic regression is used to calculate propensity scores, balancing treatment(exposure to social determinants) and control groups to minimize confounding bias. Propensity score matching ensures group comparability, reducing selection bias and enhancing the validity of causal inferences.
5) Causal Effect Estimation: Ordinary Least Squares (OLS) regression estimates treatment effects, and bootstrap sampling calculates the Average Treatment Effect (ATE) for robust causal effect estimation while managing the

bias-variance trade-off through model parameter adjustments. Results are presented in terms of ATE and 95% confidence intervals, providing a comprehensive view of causal relationships (Table III, Fig.2).

*E. Negative Controls*

By using negative controls to test variables that should not impact the outcome, we identified potential biases and unmeasured confounders[18], thereby validating the method as a useful approach for checking confounding and ensuring the model's internal consistency.Our experts designed two Negative Control Determinants (NCD): Engagement in Healthy Activities(EHA) and Non-use or short-term use of prescription opioids(NSTOP)(Supplementary Document 1[1]). We implemented these negative controls to ensure they have no impact on the outcome, OUD.Here are the detailed steps:

1) Data Preprocessing and Embedding Generation: The dataset is loaded and balanced for negative_control_opioid_use, with BioClinicalBERT embeddings (768 dimensions) used as features.
2) Subgroup Discovery: A Siamese Neural Network(SNN) identifies subgroups based on treatment with strict criteria.
3) Subsampling Identified Subgroup: After identifying subgroups, subsampling creates a balanced dataset, ensuring representative treated and untreated samples for detailed analysis.
4) Propensity Score Calculation and Matching: Logistic regression calculates propensity scores for negative_control_opioid_use to balance treatment and control groups. This helps in reducing selection bias, particularly for the negative_control_opioid_use variable, allowing for a fair comparison between groups.
5) Causal Effect Estimation: ATE is estimated using OLS regression, with bootstrap sampling for 95% confidence intervals(Table IV).

## IV. DISCUSSION

In our analysis, we compared two methodologies for calculating propensity scores and the Average Treatment Effect (ATE) of SDMHs on OUD. The Advanced Method(With SNN based Subgroup Discovery): uses Bio_ClinicalBERT embeddings with Siamese Neural Networks(SNN) and subgroup discovery, capturing semantic context and identifying meaningful subgroups, leading to more extreme ATE values and robust statistical significance (Table III). In contrast, the standard Method (Without SNN based Subgroup Discovery) relies solely on Bio_ClinicalBERT embeddings without SNN-based subgroup discovery, resulting in less extreme ATE values and potentially missing critical subgroup effects (Table III).Below is our detailed analysis.

Socially Detached (SD): It refers to individuals lacking regular interaction with family, friends, or community groups, which can stem from physical isolation, emotional disconnection, or societal exclusion. The correlation coefficient for SD is -0.02, indicating a weak negative relationship with opioid use disorder (OUD). Using a Siamese Neural Network (SNN) for subgroup discovery, the Average Treatment Effect (ATE) for

| SDMHs | Correlation Coefficient | Without SNN | | | | | With SNN | | | | |
|---|---|---|---|---|---|---|---|---|---|---|---|
| | | Treated Sample Size | Untreated Sample Size | ATE | 95% CI Lower | 95% CI Upper | Treated Sample Size | Untreated Sample Size | ATE | 95% CI Lower | 95% CI Upper |
| SD | -0.02 | 3 | 9997 | 0.007094 | 0.007094 | 0.007094 | 4985 | 5015 | -0.015866* | -0.0185 | -0.0132 |
| HCH | -0.47* | 9905 | 95 | 0.032744* | 0.032744 | 0.032744 | 4980 | 5020 | -0.483886* | -0.493099 | -0.470197 |
| OMC | 0.29* | 27 | 9973 | 0.007259* | 0.007259 | 0.007259 | 5100 | 4900 | 0.413967* | 0.322766 | 0.4333 |
| FU | -0.01 | 83 | 9917 | 0.00601* | 0.00601 | 0.00601 | 5034 | 4966 | 0.187476* | 0.179297 | 0.196402 |
| RI | 0.01 | 4966 | 5034 | -0.0010463 | -0.0571512 | 0.05625478 | 4984 | 5016 | 0.045997* | 0.040655 | 0.051002 |
| NS | -0.02 | 1 | 9999 | 0.008548 | 0.008548 | 0.008548 | 5015 | 4985 | -0.015945* | -0.0184 | -0.0134 |
| V | -0.02 | 48 | 9952 | 0.008842* | 0.008842 | 0.008842 | 4966 | 5034 | 0.023299* | 0.018597 | 0.027902 |
| JO | 0.05 | 4503 | 5497 | 0.016091* | 0.016091 | 0.016091 | 4967 | 5033 | -0.027919* | -0.0314 | -0.024297 |
| SMI | -0.10* | 7735 | 2265 | -0.017339* | -0.017339 | -0.017339 | 4964 | 5036 | 0.021832* | 0.019097 | 0.024802 |
| MDS | 0.03 | 9910 | 90 | 0.008605* | 0.008605 | 0.008605 | 5045 | 4955 | -0.036953* | -0.0419 | -0.0319 |
| AP | 0.18* | 8569 | 1431 | 0.071611* | 0.071611 | 0.071611 | 5017 | 4983 | -0.043365* | -0.0485 | -0.038397 |
| MD | 0.34* | 2372 | 7628 | -0.017337* | -0.017337 | -0.017337 | 5095 | 4905 | 0.060525* | 0.044121 | 0.0665 |
| SM | -0.45* | 9375 | 625 | -0.015975* | -0.015975 | -0.015975 | 4984 | 5016 | -0.164915* | -0.1725 | -0.157 |

TABLE IV
RESULTS OF NEGATIVE CONTROL ANALYSIS

| NCD | Treated Sample Size | Untreated Sample Size | ATE | 95% CI Lower | 95% CI Upper |
|---|---|---|---|---|---|
| EHA | 4,985 | 5,015 | -2.98E-05 | -0.00277 | 0.00268 |
| NSTOP | 5,000 | 5,000 | 0.034945 | 0.020837 | 0.048788 |

SD was -0.015866 ($p < 0.05$), signifying a highly significant negative impact on OUD within specific subgroups. This suggests that socially isolated individuals in these subgroups are less likely to misuse opioids. Without SNN, the ATE was 0.007094 ($p > 0.05$), showing no significant overall effect of SD on OUD when subgroup variations are not considered.

Health Care Handover (HCH): It refers to the transition of patients between healthcare providers to ensure continuity of care. The correlation coefficient for HCH is -0.47, indicating a strong negative relationship with opioid use disorder (OUD), suggesting that effective transitions significantly reduce OUD risk. Using SNN-based subgroup discovery, the ATE for HCH was -0.483886 ($p < 0.05$), showing a highly significant negative impact on OUD, indicating smoother handovers reduce opioid misuse. Without SNN, the ATE was 0.032744 ($p < 0.05$), which was significant, though with a smaller effect size. However, further subgroup analysis using SNN is needed to uncover more nuanced impacts of healthcare transitions on OUD progression.

Obstacles to Medical Care (OMC): It include barriers such as financial constraints, lack of transportation, and inadequate healthcare infrastructure, limiting access to services. The correlation coefficient for OMC is 0.29, indicating a moderate positive relationship with opioid use disorder (OUD), suggesting that barriers increase the risk of OUD. With SNN-based subgroup discovery, the ATE for OMC was 0.413967 ($p < 0.05$), showing a highly significant positive impact on OUD. This suggests that within specific subgroups, obstacles to medical care substantially raise the risk of OUD, likely due to difficulties in receiving timely treatment and follow-up care. Without SNN, the ATE was 0.007259 ($p < 0.05$), still significant but with a much smaller effect size, demonstrating

that subgroup analysis reveals a stronger impact on OUD risk.

Financial Uncertainty (FU): It refers to economic instability caused by unpredictable income, job loss, or insufficient resources. The correlation coefficient for FU is -0.01, indicating a very weak negative relationship with opioid use disorder (OUD), suggesting that FU may not be a major factor for OUD in the general population. Using SNN-based subgroup discovery, the ATE for FU was 0.187476 ($p < 0.05$), showing a highly significant positive impact on OUD within specific subgroups, likely due to the stress associated with financial instability. Without SNN, the ATE was 0.00601 ($p < 0.05$), also significant but with a smaller effect, underscoring the importance of subgroup analysis

Residential Instability (RI): It refers to frequent moves, homelessness, or unstable housing conditions. The correlation coefficient for RI is 0.01, indicating a very weak positive relationship with OUD, suggesting that RI may not have a substantial impact on OUD in the general population. Using SNN-based subgroup discovery, the ATE for RI was 0.045997 ($p < 0.05$), showing a highly significant positive impact on OUD, indicating that within specific subgroups, RI significantly increases the risk of OUD, likely due to stress and instability. Without SNN, the ATE was -0.0010463 ($p > 0.05$), not significant, underscoring the importance of subgroup analysis.

Nutritional Shortage (NS): It refers to a lack of access to balanced food, leading to poor nutrition. The correlation coefficient for NS is -0.02, indicating a weak negative relationship with OUD, suggesting NS may not significantly impact OUD in the general population. Using SNN-based subgroup discovery, the ATE for NS was -0.015945 ($p < 0.05$), showing a significant negative impact on OUD, implying that within specific subgroups, NS reduces the likelihood of opioid misuse possibly due to reduced access to opioid-prevalent environments. Without SNN, the ATE was 0.008548 ($p > 0.05$), not significant, highlighting the importance of subgroup analysis.

Violence (V): It refers to exposure to physical, emotional, or psychological harm, causing trauma and mental health issues. The correlation coefficient for V is -0.02, indicating a weak negative relationship with OUD, suggesting that violence

may not significantly impact OUD in the general population. Using SNN-based subgroup discovery, the ATE for V was 0.023299 ($p < 0.05$), showing a significant positive impact on OUD, suggesting that violence increases OUD risk in specific subgroups due to trauma and stress. Without SNN, the ATE was 0.008842 ($p < 0.05$), significant but with a smaller effect, highlighting the need for subgroup analysis.

Judicial Obstacles (JO): JO, such as criminal justice involvement, impact healthcare access and well-being. The correlation coefficient for JO is 0.05, indicating a weak positive relationship with OUD, suggesting JO may not significantly affect OUD in the general population. Using SNN-based subgroup discovery, the ATE for JO was -0.027919 ($p < 0.05$), showing a significant negative impact, suggesting legal deterrents may reduce OUD in specific subgroups. Without SNN, the ATE was 0.016091 ($p < 0.05$), significant but with an opposite effect, highlighting the complexity of the relationship and the importance of subgroup analysis.

Substance Misuse (SMI): It refers to the harmful use of substances like alcohol or drugs, leading to health and social issues. The correlation coefficient for SMI is -0.10, indicating a moderate negative relationship with OUD, suggesting SMI is associated with a lower likelihood of OUD. However, with SNN-based subgroup discovery, the ATE for SMI was 0.021832 ($p < 0.05$), showing a highly significant positive impact, suggesting substance misuse increases OUD risk in specific subgroups likely due to overlapping addictive behaviors. Without SNN, the ATE was -0.017339 ($p < 0.05$), significant but with an opposite effect, emphasizing the importance of subgroup analysis.

Mental Disturbance Symptoms (MDS): It includes depression, anxiety, and other psychological issues. The correlation coefficient for MDS is 0.03, indicating a weak positive relationship with OUD, suggesting MDS may not have a substantial impact on OUD in the general population. Using SNN-based subgroup discovery, the ATE for MDS was -0.036953 ($p < 0.05$), showing a highly significant negative impact, suggesting that individuals with MDS in specific subgroups are less likely to misuse opioids, possibly due to seeking professional help. Without SNN, the ATE was 0.008605 ($p < 0.05$), significant but with an opposite effect, emphasizing the importance of subgroup analysis.

Acute Pain (AP): It refers to severe, sudden pain from injury, surgery, or medical conditions, often requiring short-term management. The correlation coefficient for AP is 0.18, indicating a moderate positive relationship with OUD, suggesting that AP is linked to a higher likelihood of OUD. Using SNN-based subgroup discovery, the ATE for AP was -0.043365 ($p < 0.05$), showing a highly significant negative impact, indicating that within specific subgroups, individuals with acute pain are less likely to misuse opioids, likely due to proper pain management. Without SNN, the ATE was 0.071611 ($p < 0.05$), significant but with an opposite effect, highlighting the importance of subgroup analysis.

Medical Disability (MD): It refers to chronic health conditions that impair daily activities and require ongoing care. The correlation coefficient for MD is 0.34, indicating a strong positive relationship with OUD, suggesting that MD increases

the likelihood of OUD. Using SNN-based subgroup discovery, the ATE for MD was 0.060525 ($p < 0.05$), showing a highly significant positive impact, implying that individuals with medical disabilities in specific subgroups are more likely to misuse opioids, likely due to chronic pain and limited mobility. Without SNN, the ATE was -0.017337 ($p < 0.05$), significant but with an opposite effect, underscoring the complexity of the relationship and the need for subgroup analysis.

Suicide Mortality (SM): It refers to the rate of deaths caused by self-inflicted harm, reflecting severe mental health crises. The correlation coefficient for SM is -0.45, indicating a strong negative relationship with OUD, suggesting that higher suicide mortality rates are associated with lower OUD rates. Using SNN-based subgroup discovery, the ATE for SM was -0.164915 ($p < 0.05$), showing a highly significant negative impact, implying that in specific subgroups, higher suicide mortality is linked to lower OUD, possibly due to overlapping risk factors. Without SNN, the ATE was -0.015975 ($p < 0.05$), also significant but with a smaller effect, emphasizing the need for subgroup analysis.

Furthermore, our negative control result analysis aligns with expectations, showing minimal ATEs and confidence intervals close to zero for EHA and NSTOP, indicating no significant impact on OUD. This validates the model's ability to correctly identify factors with no real association with OUD, reinforcing the reliability of the causal inference results. These findings confirm that EHA and NSTOP are appropriate negative controls and ensure there are no biases or unmeasured confounders affecting the results.

### A. Key Findings

Our key findings of Social Determinants of Mental Health (SDMHs) and their impact on OUD highlight several key findings. Significant determinants include healthcare handover, obstacles to medical care, financial uncertainty, residential instability, nutritional shortage, violence, judicial obstacles, substance misuse, mental disturbance symptoms, acute pain, medical disability, and suicide mortality. These determinants provide crucial areas for targeted interventions to address and mitigate OUD.

Correlation analysis revealed several significant relationships between SDMHs and OUD. For instance, healthcare handover (HCH) had a strong negative correlation (-0.47), indicating that effective healthcare transitions reduce OUD rates. Obstacles to medical care (OMC) showed a positive correlation (0.29), suggesting that barriers to accessing care increase the likelihood of OUD. Residential instability (RI) had a weak positive correlation (0.01), indicating a slight association with higher OUD rates. However, correlation alone is insufficient to establish causation, as it shows relationships between variables without accounting for confounding factors or the directionality of cause-and-effect. Thus, Causation analysis is necessary to determine the true impact of these determinants on OUD and design effective interventions.

Using SNN-based subgroup discovery, we identified significant causal effects of social determinants on OUD. For example, the ATE for healthcare handover (HCH) was -0.483886 ($p < 0.05$), highlighting the importance of effective healthcare

transitions. The ATE for obstacles to medical care (OMC) was 0.413967 ($p < 0.05$), indicating a highly significant positive impact on OUD, suggesting that reducing obstacles to medical care can significantly lower the risk of OUD within certain subgroups. Financial uncertainty (FU) showed a highly significant positive impact on OUD within specific subgroups with an ATE of 0.187476 ($p < 0.05$), highlighting the stress and instability associated with financial struggles.

SNN-based subgroup discovery is preferred because it accounts for heterogeneity within the population and identifies specific subgroups where the causal effects are more pronounced. This approach provides a more nuanced understanding of how social determinants impact OUD, allowing for targeted interventions. For instance, while the correlation analysis suggested a weak relationship between financial uncertainty and OUD, the SNN-based analysis revealed a stronger and more significant causal effect within specific subgroups (ATE = 0.187476, $p < 0.05$).

Without SNN-based subgroup discovery, the causal effect estimates can be misleading. For example, the ATE for Socially Detached (SD) without SNN was 0.007094 ($p > 0.05$), suggesting an insignificant effect, which contrasts sharply with the significant negative effect identified with SNN.Similarly, the ATE for obstacles to medical care (OMC) without SNN was 0.007259 ($p < 0.05$), but the effect size was much smaller compared to the significant impact identified with SNN. These discrepancies underscore the limitations of overlooking subgroup variations, which can lead to inaccurate conclusions about the impact of social determinants on OUD.

Alternatively, we employed a negative control to validate the model's internal consistency and control for confounding variables. While this approach ensures robustness within the current dataset, generalizing the model across diverse populations remains a crucial next step. Given the time constraints and intensive effort required for dataset annotation, broader validation was beyond the scope of this study.Future work will focus on conducting evaluations on additional datasets and extending the framework to other populations, such as cancer patients, to further assess its adaptability, generalizability, and efficacy in varied clinical settings.

Our analysis underscores the importance of understanding both correlation and causation when examining the impact of social determinants on OUD. While correlation provides initial insights, causation analysis with SNN-based subgroup discovery offers a more accurate understanding, identifying subgroups where the impact is most significant. This guides more effective and targeted interventions to mitigate OUD. Our study highlights key determinants that inform clinical interventions, improve healthcare transitions, and reduce care barriers. By efficiently identifying risk factors from clinical notes, clinicians can develop personalized treatment plans. The findings also guide public health policies, focusing on vulnerable populations through targeted interventions and resource allocation, ultimately improving patient outcomes and reducing the societal burden of OUD. Furthermore, the framework we developed for OUD is adaptable to other disorders influenced by social determinants, with necessary modifications such as customizing determinants, retraining models, redefining outcome variables, and validating on relevant datasets, offering insights across a range of conditions.

## V. CONCLUSION

In this study, we applied multiple causal inference methods with Health embeddings to investigate the effects of social determinants of mental health on Opioid Use Disorder (OUD) outcomes. Using advanced techniques, including the Clinical-Longformer model and Subgroup Discovery with Siamese Neural Networks(SNN), we provided robust causal estimates, highlighting the importance of mental health issues and chronic conditions in OUD. Our findings reveal that certain determinants increase the risk of OUD complications, while others are associated with a lower risk of hospitalization. Identifying both detrimental and beneficial effects allows us to guide preventive care and propose repurposing of determinants for OUD treatment. These results emphasize the need for targeted interventions, such as improving healthcare transitions and mental health support, to effectively mitigate OUD and inform public health strategies.

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
