# OpenReview forum: "Finding Impacts of Social Determinants of Mental Health on Opioid Use Disorder Progression from Clinical Notes Using a Siamese Neural Network-Based Causal Effect Model"
_IEEE.org/EMBS/BHI/2024/Conference — IEEE BHI'24_

### Official Review · Reviewer_7Hot · 2024-08-10
**Two-Step framework for identifying Effects of Social Determinants on OUD Progression**

**Overall Rating:** 6
**Confidence:** 4

**Other Quality Metrics:**

Clarity of Writing: Fair
Scientific Rigor: Good
Methodological Completeness: Good
Experiments and Results: Great

**Questions For The Authors:**

1. The title could be simplified.
2. The SDMH-OUD-Clinic dataset you constructed is extracted from the patient electronic health records in the MIMIC-IV dataset. Do these electronic records include the information of 13 social determinants defined in the manuscript?

**Strengths:**

The manuscript presents a novel framework for identifying the impact of social determinants on OUD progression. The manuscript is well-structured, with robust and credible experimental results, and the authors have provided a comprehensive analysis of these results. The findings offer valuable insights for the development of targeted public health strategies, which can inform specific measures to mitigate the impact of OUD and have significant practical applications.

**Summary Of The Paper:**

This manuscript focuses on identifying the causal effects of social determinants on the progression of Opioid Use Disorder (OUD). The authors propose a two-step framework: first, they use a Multitask Multilabel Clinical-Longformer model to detect and classify social determinants of mental health from clinical notes; second, they employ a Siamese Neural Network-based method to discover causal relationships within identified subgroups of patients. The study uses a newly created dataset, SDMH-OUD-Clinic, derived from MIMIC-IV data, and validates the models on this dataset as well as on the Infant Health and Development Program(IHDP) dataset. The results indicate the effectiveness of the proposed methods in providing detailed causal explanations, which could inform targeted strategies to mitigate the impact of OUD based on specific social determinants.

**Weaknesses:**

The comprehensiveness of the dataset that was constructed may potentially impact the experimental results.

---

### Official Review · Reviewer_CRxB · 2024-08-10
**Finding Impacts of Social Determinants of Mental Health on Opioid Use Disorder Progression from Clinical Notes Using a Siamese Neural Network-Based Causal Effect Model: A Review**

**Overall Rating:** 6
**Confidence:** 3

**Other Quality Metrics:**

1. Clarity of Writing: Good
2. Clinical Significance: Good
3. Methodological Novelty: Good
3. Experiments and Results: Great

**Questions For The Authors:**

1. How do you envision the integration of your findings into clinical decision-making processes? Are there specific interventions or policy recommendations that could be derived from your work?
2. Can you elaborate on the challenges faced when creating the SDMH-OUD-Clinic dataset, particularly in terms of annotation consistency and the handling of missing data?
3. The study focuses on OUD, but could the same framework be applied to other disorders or conditions influenced by social determinants? If so, what modifications would be necessary?

**Strengths:**

1. Innovative Methodology: The paper introduces a novel combination of MMCL and SNN models, which effectively leverage clinical text data to identify causal relationships in a complex and high-impact domain.
2. Dataset Contribution: The creation of the SDMH-OUD-Clinic dataset, annotated for SDMHs and OUD, represents a valuable contribution to the research community, enabling further study in this area.
3. Comprehensive Evaluation: The study employs robust evaluation metrics and compares the proposed model against several baseline methods, demonstrating the superiority of their approach.

**Summary Of The Paper:**

The paper presents a study focused on understanding the causal effects of social determinants of mental health (SDMHs) on the progression of Opioid Use Disorder (OUD). The authors propose a novel two-step framework involving a Multitask Multilabel Clinical-Longformer (MMCL) model to detect relevant social determinants from clinical notes and a Siamese Neural Network-based (SNN) subgroup discovery technique to ascertain their causal effects. The study also introduces the SDMH-OUD-Clinic dataset, created from clinical notes annotated for SDMHs and OUD, and evaluates the proposed models using both this dataset and the existing IHDP dataset. The results demonstrate the effectiveness of the models in identifying and explaining causal relationships between social determinants and OUD.

**Weaknesses:**

1. Complexity in Interpretation: While the methods used are sophisticated, the complexity of the approach may limit the accessibility of the findings to a broader audience, particularly those without a strong background in machine learning and causal inference.
2. Limited Clinical Application: The study primarily focuses on the identification of causal relationships but offers limited discussion on how these findings could be translated into actionable interventions or policies in clinical practice.
3. Generalization: The results, while promising, are based on specific datasets. There is a need for further validation across different populations and clinical settings to ensure generalization.

---

### Official Review · Reviewer_MzP8 · 2024-08-28
**Finding Impacts of Social Determinants of Mental Health on Opioid Use Disorder Progression from Clinical Notes Using a Siamese Neural Network-Based Causal Effect Model**

**Overall Rating:** 7
**Confidence:** 4

**Other Quality Metrics:**

a) Clarity of writing; excellent
(b) Clinical Significance; fair
(c) Methodological Novelty; great
(d) Experiments and Results: good

**Questions For The Authors:**

1) Have you checked if the traditional transformer model trained on clinical dataset performs worse than longformer in this case?
2) Can you please give us an estimate of the average length of the clinical texts used in the dataset?
3) Can you please explain the cleaning methods that you used to clean the text data from discharge.csv file?

**Strengths:**

1) Their approach of using Longformer at the beginning to detect determinants of OUD and using siamese neural network with shared representation learning suit good for this particular problem. So, in terms of methods, I think their method is sound and novel.

2) Their siamese neural network model outperforms other approaches for both in sample and out sample cases for IHDP dataset.

3) The effect of social determinants on OUD is an interesting problem, having good implication for this clinical field. They found significant  correlations between some of these determinants and OUD.

**Summary Of The Paper:**

The authors used Clinical Longformer model to identify 13 social determinants of Mental health from clinical notes. These 13 determinants are well established categories from prior work. First of all, the model does a binary prediction on the presence or absence of  OUD, and if OUD is present, the model further identifies the social determinants of mental health. At the next step, they used Siamese Neural Network to find the casual effects of these determinants on OUD. The Siamese Neural Network based model showed better performance in terms of PEHE scores for in sample and out sample cases for IHDP dataset. They noted the correlation of each of the 13 determinants of mental health  with OUD and found significant correlation for some of the determinants with OUD which is interesting.

**Weaknesses:**

1) Using more dataset for evaluation would be great to show that this model is generalizable across different datasets.

2)I understand that using longformer is good for long text cases. I think the average length of clinical texts for the dataset should be mentioned to further demonstrate the use of longformer in this case, otherwise, it  will increase the model parameters unnecessarily. Have you checked if the traditional transformer model trained on clinical dataset performs worse than longformer in this case?

---

### Decision · Program_Chairs · 2024-09-23

Accept